⊕ | Open Peer Review | Bacteriology | Research Article

# Gut commensal *Alistipes shahii* improves experimental colitis in mice with reduced intestinal epithelial damage and cytokine secretion

Xiaoying Lin,[1,2] Mingchao Xu,[3,4] Ruiting Lan,[5] Dalong Hu,[5] Suping Zhang,[2] Shuwei Zhang,[1,2] Yao Lu,[2] Hui Sun,[2] Jing Yang,[2,6] Liyun Liu,[2,6,7] Jianguo Xu[1,2,6]

**ABSTRACT** The commensal bacterium *Alistipes shahii* is a core microbe of the human gut microbiome and its abundance is negatively correlated with inflammatory bowel diseases (IBDs). However, its fundamental role in regulating inflammatory response remains unknown. Using a dextran sulfate sodium (DSS)-induced colitis mouse model, we examined the effect of *A. shahii* strain As360 intervention on host inflammatory response and found that *A. shahii* As360 alleviated disease activity index, colon shortening, and colonic histopathological lesion. The levels of tight junction proteins (mainly ZO1 and claudin-1) were decreased in DSS-induced colitis mice, whereas the levels of these proteins were elevated in colitis mice with *A. shahii* As360 treatment. In addition, *A. shahii* As360 treatment led to alterations in cytokine release, especially an increase of IL10. It also led to reduced expressions of *mtor* and *Nlrp3* and increased expression of mTOR inhibitor *Ddit4* at the transcriptional level. 16S rRNA amplicon sequencing found that *Bacteroides*, a producer of short-chain fatty acids (SCFAs), was enriched in the fecal samples of mice with *A. shahii* treatment. Metabolic analyses found that, following *A. shahii* As360 treatment, the SCFAs in the fecal content was increased whereas lactic acid was decreased in the cecal content. These findings suggest that supplementation with *A. shahii* As360 is a promising strategy to prevent colitis.

**IMPORTANCE** As one of the core microbes and keystone species in the human gut, *Alistipes shahii* has the potential to inhibit inflammation and improve inflammatory bowel diseases (IBDs) conditions. In this study, we experimentally demonstrated that oral administration of *A. shahii* As360 alleviated symptoms of colitis, altered the release of cellular inflammatory factors, reduced the intestinal epithelial barrier damage, and changed gut microbiota and fecal metabolites. These findings provide a deeper understanding of the beneficial effects of *A. shahii* and its perspective for better strategies to prevent IBD.

**KEYWORDS** *Alistipes shahii*, inflammatory bowel diseases, short-chain fatty acids, inflammatory cytokines, mTOR

G ut microbiota is widely involved in the basic physiological activities of the host (1). Its dysbiosis, especially a loss of beneficial symbionts, may result in aggravated mucosal damage, translocation of symbiotic microorganisms, and aberrant immune activation, thus triggering chronic inflammation (2). The intestinal commensal bacterium *Alistipes shahii* has been defined as one of the core microbes and keystone species in human gut microbiota. Its abundance varies and may be correlated with gut diseases (3). Based on metagenomic studies, several *Alistipes* species, including *A. shahii*, were depleted in the gut of inflammatory bowel diseases (IBDs) patients (4). A follow-up study

**Peer Reviewers** Shuai Hao, Beijing Technology and Business University, Beijing, China; Ruifu Yang, Beijing Institute of Microbiology and Epidemiology, Beijing, China

Address correspondence to Liyun Liu, liuliyun@icdc.cn, or Jianguo Xu, xujianguo@icdc.cn.

Xiaoying Lin, Mingchao Xu, and Ruiting Lan contributed equally to this article. The author order was determined by their contribution to the article.

The authors declare no conflict of interest.

of 132 subjects and 2,965 biological samples (stool, biopsy, and blood specimens) also showed a low relative abundance of *A. shahii* in the two most prevalent forms of IBD, dysbiotic ulcerative colitis (UC) and Crohn's disease (CD) (5). Moreover, the abundance of *A. shahii* has been found to be negatively correlated with CD activity (6).

IBD is a chronic condition of immune-mediated intestinal inflammation driven by both genetic predisposition and environmental factors (7), affecting 0.3–0.5% of the global population (8, 9). There are always non-negligible changes in the quantity or products of gut microbiota in patients with IBD (10), that cause the activation of immune cells and a series of receptors as well as an overload of cytokines (11), ultimately exposing the intestinal barrier to danger. The current treatment for IBD, includes immunomodulators, biologics, and monoclonal antibodies like infliximab (12), mostly designed to suppress the host immune system rather than directly regulate the gut microecosystems to relieve inflammation. Therefore, replenishing the significantly reduced beneficial bacteria in the gut of IBD patients may become a new strategy to improve the inflammatory status (13). For example, *Faecalibacterium prausnitzii* was found to be significantly lower in abundance in UC patients (14). Supplementation of *F. prausnitzii* significantly ameliorated DSS-induced colitis at the level of clinical symptoms, histological inflammation, and immune status (15).

Recent studies have found that many beneficial commensal bacteria or probiotics exert anti-inflammatory effects through short-chain fatty acids (SCFAs) (16). SCFAs, mainly acetic, propionic, and butyric acids (17), can activate G protein-coupled receptor 43 to maintain intestinal mucosal integrity and barrier stability (18, 19) and create a localized acidic environment in the intestine to inhibit pathogen proliferation (20, 21). *A. shahii* is an SCFAs producer and can metabolize sugar to succinic, acetic, and propionic acids (22). We hypothesized that *A. shahii* as a core gut microbe and a producer of SCFAs has the potential to be used for the therapy of IBD. Therefore, the primary aim of this study was to examine the anti-inflammatory efficacy and mechanism of *A. shahii* using an experimental mouse model of colitis and determine whether intervention with *A. shahii* could ameliorate experimental colitis in mice.

## RESULTS

### Antibiotic susceptibility of *A. shahii* strains

Four *A. shahii* strains were isolated from fecal samples of healthy individuals. These strains were tested for susceptibility to 10 antibiotics (Table S1). As360 was susceptible to the 10 antibiotics tested, while the other three strains were resistant to two to seven of the drugs tested including clindamycin and moxifloxacin. Given the risk of horizontal transfer of antibiotic resistance genes to other species in the gut (23), strain As360 was selected for further experiments due to its favorable susceptibility profile.

### *A. shahii* treatment alleviated the symptoms of DSS-induced colitis

To determine the anti-inflammatory effect of *A. shahii*, strain As360 was used to intervene in dextran sulfate sodium (DSS)-induced colitis model mice (Fig. 1A). The mice were divided into three groups: (i) mice without DSS treatment as the normal control (NC) group, (ii) DSS treatment only as the DSS group; and (iii) DSS treatment plus *A. shahii* As360 intervention as the As360 group. During the 7-day pre-gavage stage, the body weight of mice was comparable among the three groups (Fig. S1). From day 5 of the modeling phase, the body weight in the DSS-induced colitis mice was significantly lower than that in the NC mice ($P < 0.05$), and intervention with *A. shahii* As360 exhibited an increasing trend ($P > 0.05$, Fig. 1B). On days 3–6 of DSS exposure, the disease activity index (DAI) scores in the DSS group were significantly higher than those in the NC group, while administration of *A. shahii* As360 decreased the DAI scores ($P < 0.05$ vs the DSS group) on days 5 and 6 (Fig. 1E). Compared with the NC group, colon length was notably shortened in the DSS group ($P < 0.001$), and colonic shortening was less pronounced in the As360 group than in the DSS group ($P < 0.001$, Fig. 1C and D).

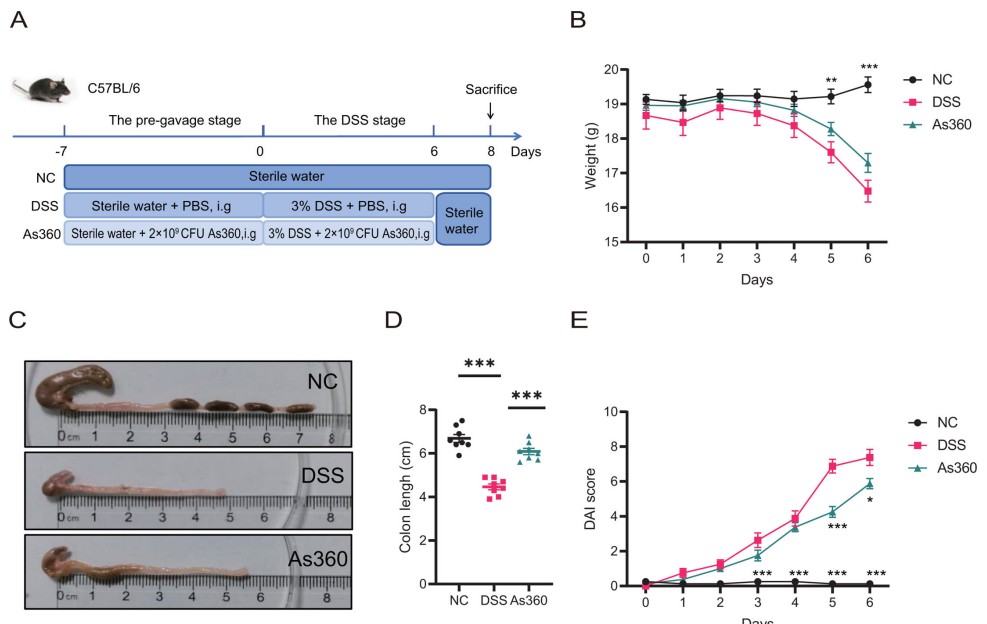

**FIG 1** Effect of *A. shahii* on severity of DSS-induced colitis in mice. (A) Experiment design pattern diagram. (B) Weight change for 7 days in the DSS stage. (C) Representative images of colons in three group mice. (D) Quantification of the colon length in four group mice. (E) DAI scores of three group mice in DSS stage. Statistical comparison was performed by a one-way ANOVA followed by Dunnett's multiple comparisons test. Values represent mean ± standard error. Number of mice per group, *n* = 8; *$P < 0.05$, **$P < 0.01$, and ***$P < 0.001$ indicated the significant differences of the DSS versus NC or As360 group. As360: DSS-induced mice treated with *A. shahii* As360; DSS: DSS-induced mice treated with phosphate buffer saline; NC: normal control mice.

## *A. shahii* treatment ameliorated colonic damage in mice with colitis

Microscopic observation of colon histopathological sections revealed that DSS-induced histopathological damage was generally moderate to severe, and mucosal lesions and submucosal inflammatory cell infiltration were observed in some visual fields. In the *A. shahii* As360 group, histopathological scores were significantly reduced, although accompanied by slight histopathological damage ($P < 0.05$, Fig. 2A and B). Moreover, the mRNA expression of *Muc2* was significantly lower in the DSS group than in the NC and As360 groups ($P < 0.05$, Fig. 2D).

To clarify changes in the mucosal barrier, we determined the expressions of tight junction proteins, zonula occludens 1 (ZO1), claudin-1, and occludin, in the three groups. Compared with the NC group, the mRNA expressions of *Zo1*, claudin 1, and occludin in the DSS group were significantly decreased, while *A. shahii* As360 intervention reversed this trend ($P < 0.05$, Fig. 2C). Immunostaining of ZO1 and claudin-1 in distal colon sections showed higher protein expressions in the As360 group than in the DSS group (Fig. 3).

## *A. shahii* treatment modulated local and systemic inflammation of colitis

As shown in Fig. 4A, compared with the NC group, exposing the mice to DSS significantly increased the levels of interleukin 1 beta (IL1β), tumor necrosis factor alpha (TNFα), and interleukin 6 (IL6) as well as significantly decreased the levels of interleukin 10 (IL10) in the colonic tissue; however, *A. shahii* As360-treated mice had higher level of IL10 and lower level of IL1β, TNFα, and IL6 than the DSS group ($P < 0.05$). Moreover, after *A. shahii* As360 intervention, DSS-induced mice had higher serum IL10 levels ($P < 0.05$, Fig. 4B). In addition, *A. shahii* As360 treatment also increased the secretion of IL10 in the lipopolysaccharide (LPS)-stimulated RAW264.7 cell inflammation model ($P < 0.05$, Fig. 4C).

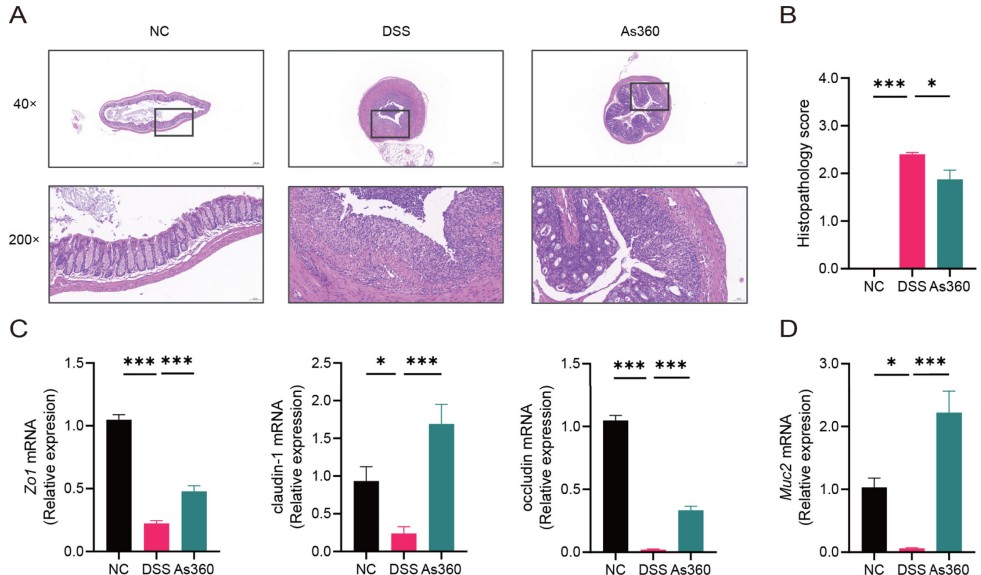

**FIG 2** Effect of the treatment with *A. shahii* on colon mucosal barrier damage in DSS mice. (A) Representative images of HE stained sections of ascending colons from indicated groups. (B) Histopathological score. Number of mice per group, n = 8. (C and D) Relative expression of *Zo1*, claudin-1, occludin, and *Muc2* in mice colons by qRT-PCR, No of mice per group n = 5. As360: DSS-induced colitis mice dosed with *A. shahii* As360 group, slight histopathological damage; DSS: DSS-induced colitis group, moderate to severe histopathological damage; NC: normal control group, no pathology. Scale bar = 50 and 200 μm. Statistical comparison was performed by a one-way ANOVA followed by Dunnett's multiple comparisons test. Values represent mean ± standard error; *P < 0.05 and ***P < 0.001.

Since IL10 inhibits the activity of mammalian target of rapamycin (mTOR) by inducting DNA damage-inducible transcript 4 (DDIT4) protein to negatively regulate NOD-like receptor thermal protein domain associated protein 3 (NLRP3) inflammasome activation (24), we used quantitative real-time polymerase chain reaction (qRT-PCR) to determine the mRNA expression of these three genes in the colon. As shown in Fig. 4D through F, compared with the DSS group, the mRNA expression of the *mtor* and *Nlrp3* genes were reduced, and the expression of mTOR inhibitor *Ddit4* gene was increased in the As360 group (P < 0.05).

## *A. shahii* treatment altered the gut microbiota of colitis mice

To elucidate the effects of *A. shahii* on gut microbiota of the DSS-induced mice, the V3–V4 region of the 16S rRNA gene was sequenced. The rarefaction curves of each sample showed that the sequencing was deep enough to reliably characterize the bacterial microbiome (Fig. S2). Compared with the NC group, ACE and Chao1 indices of the DSS group were significantly decreased (P < 0.05). In the *A. shahii* As360 group, the intestinal microbial diversity decreased less than in the DSS group (P < 0.05 in both cases; Fig. 5A).

Principal coordinates analysis (PCoA) showed that the fecal samples were clustered into three groups based on gut microbiota composition, and the samples of the As360 group were close to those of the NC group (Fig. 5B). Firmicutes and Bacteroidetes accounted for the majority (>90%) of fecal microbiota, and the abundance of Verrucomicrobiota ranked third in the As360 group (Fig. 5C). The top five genera with the highest relative abundance in the NC group were *Alistipes*, *Ligilactobacillus*, uncultured Bacteroidales bacterium, *Alloprevotella*, and *Prevotellaceae* UCG 001. In the DSS group, the relative abundance of *Ligilactobacillus* was reduced, and *Lachnospiraceae* NK4A136 group was the most dominant. The relative abundance of *Ligilactobacillus* and *Akkermansia* was increased almost one-fold in the As360 group (Fig. 5D). Linear discriminant analysis effect size (LefSe) showed that *Bacteroides*, which are known as producer of SCFAs, were enriched in the As360 group (Fig. 5E).

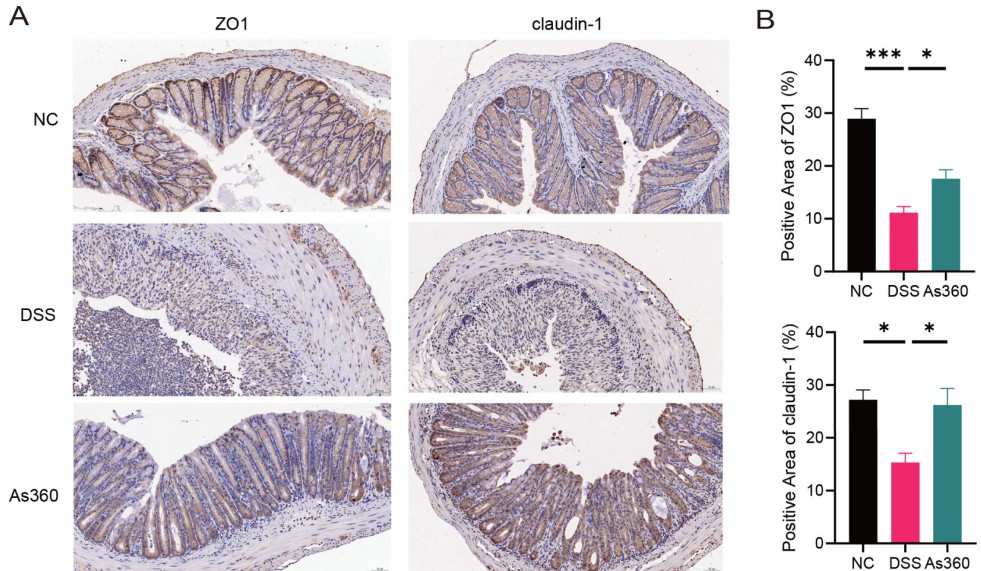

**FIG 3** Expression of tight junction proteins ZO1 and claudin-1 in the colons tested by immunohistochemical assays in *A. shahii* treated DSS mice. (A) Representative images, scale bar = 50 µm. (B) Quantification expression. Statistical comparison was performed by a one-way ANOVA followed by Dunnett's multiple comparisons test. Values represent mean ± standard error; number of mice per group $n = 5$, *$P < 0.05$.

## *A. shahii*-treated mice had higher SCFAs in feces

Targeted metabolomic profiling of mouse feces showed that the total levels of SCFAs in the As360 mice were significantly higher than those in the DSS mice, with acetic and propionic acids accounting for the major composition (Fig. 6A and B). Similarly, the fermentation supernatant of *A. shahii* As360 produced mainly acetic and propionate acids (Table S2). Notably, compared with the NC group, the mRNA expression of colonic SCFAs transporter, sodium-coupled monocarboxylate transporter (Smct), was significantly downregulated in the DSS group, while significantly upregulated in the As360 group ($P < 0.05$, Fig. 6C).

## *A. shahii* treatment altered the cecal metabolites in DSS-induced colitis mice

To generate metabolic profiles of normal and colitis mice before and after *A. shahii* treatment, we next performed cecal metabolome analysis in an untargeted manner. A total of 2,010 and 325 differential metabolites were identified between the NC and DSS groups, and between the DSS and As360 groups, respectively (Fig. 7A and B).

Enrichment analysis of the Kyoto Encyclopedia of Genes and Genomes (KEGG) pathway was performed on the differential metabolites. Histidine metabolism pathway and the pyruvate metabolism pathway were the most differentially regulated pathways between the DSS group and the NC group (Fig. 7C) and between the DSS group and the As360 group (Fig. 7D), respectively.

The level of metabolites in the fecal content related to the two pathways was further examined. The level of histidine, related to histidine metabolism, in the DSS group was significantly higher than that in the NC group (Fig. 7E). The level of lactic acid, related to the pyruvate metabolism, was significantly lower in the As360 group compared with the DSS group (Fig. 7F).

## DISCUSSION

In this study, we employed the experimental colitis mouse model to demonstrate anti-inflammatory ability of *A. shahii* and found that oral administration of *A. shahii* As360 was effective in reducing DAI scores, colon shortening, and colon tissue inflammation.

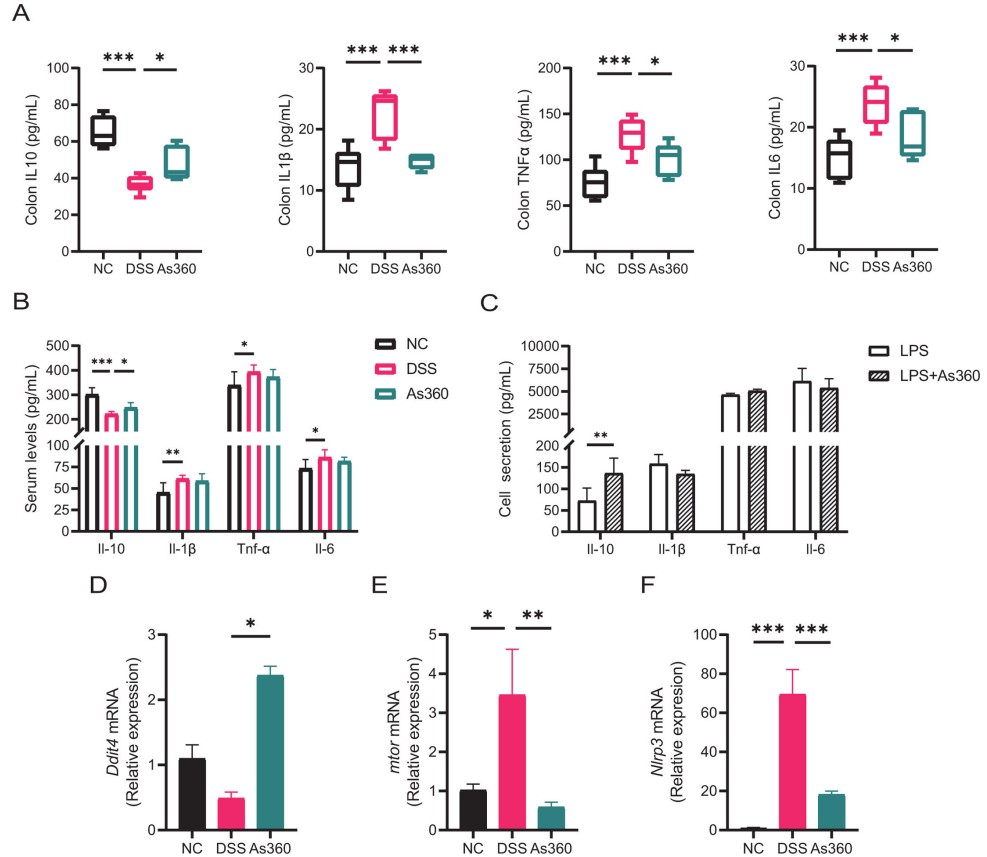

**FIG 4** Effect of *A. shahii* As360 on the secretion of cytokines and the expression of *Ddit4*, *mtor*, and *Nlrp3* genes. (A–C) Levels of cytokines in mice colons, serum, and LPS-induced RAW264.7 cells culture supernatant. Number of mice per group $n = 3$. (D–F) The relative expression of *Ddit4*, *mtor*, and *Nlrp3* in the colons. Number of mice per group, $n = 5$. Statistical comparison was performed by a one-way ANOVA followed by Dunnett's multiple comparisons test. Values represent mean ± standard error; *$P < 0.05$, **$P < 0.01$, and ***$P < 0.001$ indicated the significant differences of the DSS versus NC or As360 group *in vivo*, and the significant differences of LPS versus LPS + As360 *in vitro*. As360: DSS-induced mice treated with *A. shahii* As360; DSS: DSS-induced mice treated with phosphate buffer saline; NC: normal control mice.

Notably, the secretion and expression of anti-inflammatory factor IL10 were increased in the serum and colonic tissue of *A. shahii*-treated colitis mice. IL10 is a key anti-inflammatory cytokine produced by activating immune cells, which, to a certain extent, restricts the activation of inflammasome and inflammatory response by promoting mitochondrial autophagy, thereby preventing the development of intestinal inflammation (25, 26). We further found that in *A. shahii* As360-treated colitis mice, the expression of *mtor* and *Nlrp3* genes was reduced while the expression of mTOR inhibitor gene *Ddit4* was increased. Studies have shown that IL10 inhibited mTOR-induced mitophagy by inducing mTOR inhibitor DDIT4 and then negatively regulated the activation of NLRP3 inflammasome (24). A previous study has shown that NLRP3 is a direct binding partner for the autophagy inhibitor mTOR and in colitis mice, hypoxia ameliorated intestinal inflammation by reducing NLRP3-mTOR binding and thus activating autophagy (26). Therefore, *A. shahii* As360 modulated local and systemic inflammation to ameliorate colitis. However, the specific mechanism needs to be further elucidated.

Tight junction proteins play an important role in maintaining the mechanical barrier of the intestinal mucosa (27). Disrupted barrier function in patients with IBD is associated with reduced expression of the tight junction scaffold protein ZO1, claudin-1, and occludin (28). Our study showed that the expression of ZO1 and claudin-1 at both mRNA and protein levels and the expression of occludin at mRNA level were significantly

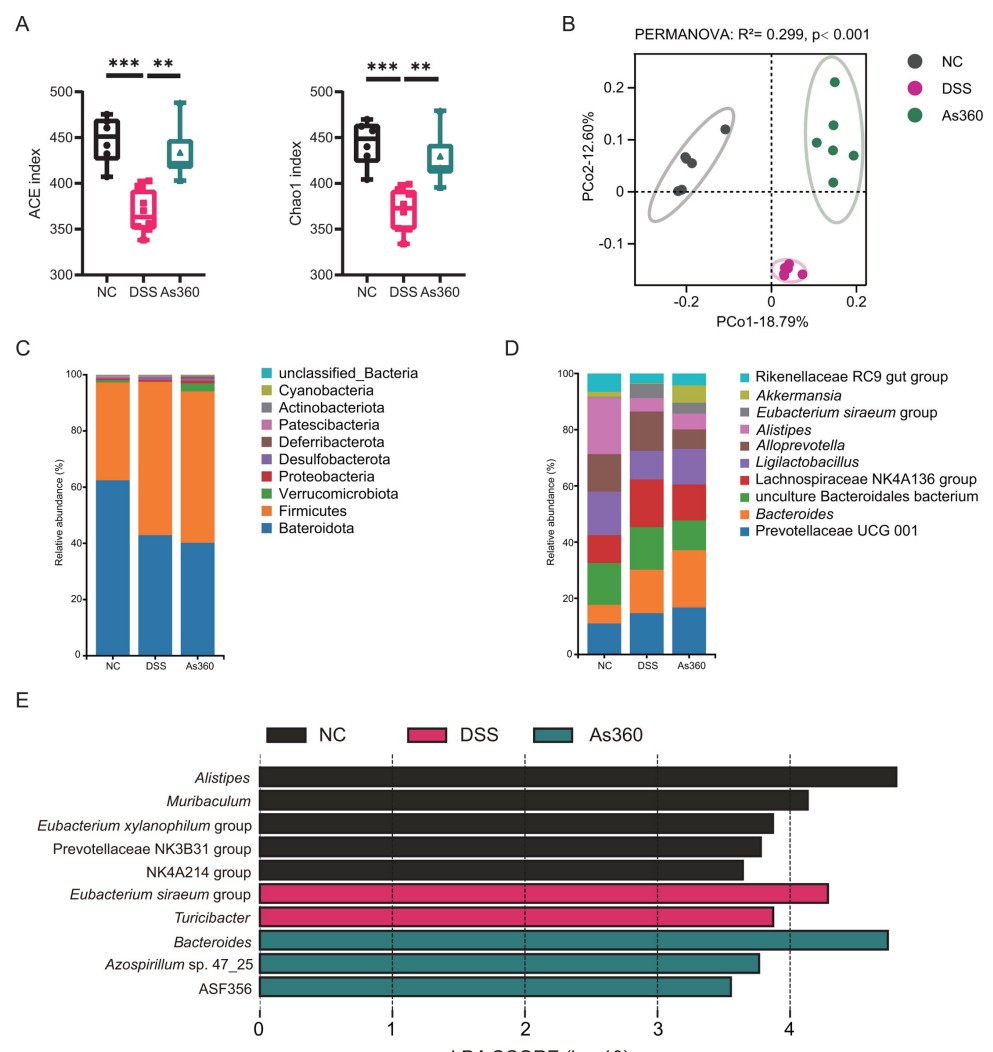

**FIG 5** Effect of *A. shahii* As360 treatment on intestinal flora diversities and structures in DSS mice. (A) Alpha diversity indices comparison among different groups. Statistical comparison was performed by a one-way ANOVA followed by Dunnett's multiple comparisons test. (B) PCoA based on the unweighted UniFrac distance. Similarity analysis was performed to test the differences between sample points of different groups. (C and D) Barplot of the microbial compositional profiles at the phylum and genus level (top 10). (E) Overrepresented bacterial taxa among groups at the genus level determined by linear discriminant analysis (LDA). Values represent mean ± standard error; number of mice per group $n = 6$, *$P < 0.05$ and **$P < 0.01$.

reduced in DSS-induced colitis mice, but they were significantly increased in *A. shahii* As360-treated colitis mice. Mucin and antimicrobial peptides secreted by goblet cells are equally important in protecting mucosal barrier integrity from bacterial degradation (29). We found that treatment with *A. shahii* significantly increased the mRNA expression of mucus gene *Muc2*, which was decreased in DSS-induced colitis mice.

A key feature of IBD is altered composition of the gut microbiota, known as dysbiosis, characterized by an overall decrease in microbial diversity with a loss of beneficial symbionts (2). Our 16S gene amplicon sequencing data showed that the microbial composition differed among the three groups. Treatment with *A. shahii* increased intestinal microbial diversity and shifted gut microbial structure perturbed by DSS closer to that of the NC group. The *A. shahii* As360 intervention group enriched highest amount of *Akkermansia* among the three groups. *Akkermansia* has been reported to play a role in the maintenance of a healthy gut barrier and limiting the onset of inflammation (30). The *A. shahii* As360 intervention group had a higher relative abundance of

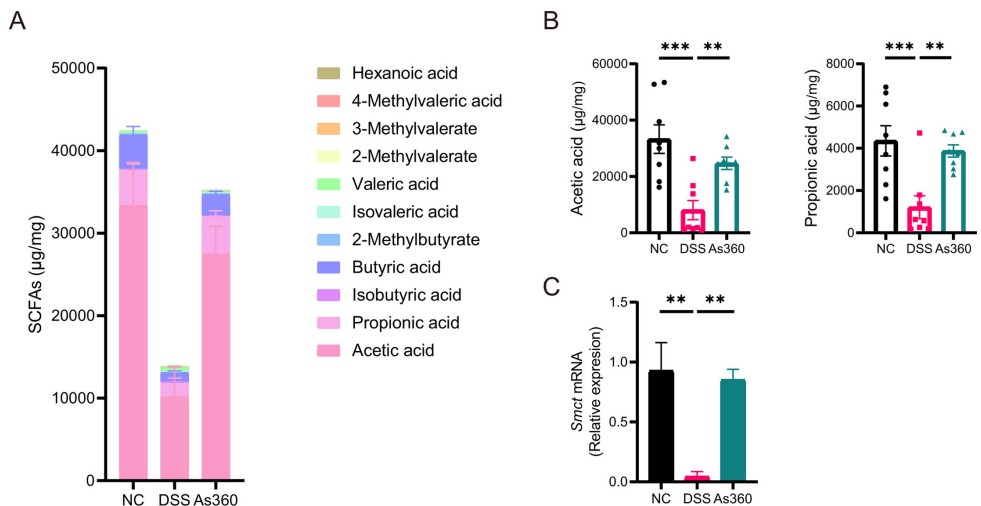

**FIG 6** Effect of *A. shahii* As360 treatment on mice fecal short-chain fatty acids (SCFAs) profiles. (A) Level of SCFAs. Number of mice per group, $n = 8$. (B) Level of acetic and propionic acids, $n = 8$. (C) Relative expression of *Smct* in the colons by qRT-PCR. Number of mice per group, $n = 5$. **$P < 0.01$ and ***$P < 0.001$.

*Ligilactobacillus* than the DSS group. Studies have shown that many bacteria in the genus *Ligilactobacillus* have beneficial effects; for example, *L. salivarius* CCFM 1266 attenuates immune checkpoint blockade-induced colitis (31) and *L. acidipiscis* YJ5 modulates the gut microbiota and enhances the mucosal barrier (32). Besides, according to LefSe analysis, we found more potential SCFAs producers in the As360 group, like *Bacteroides*. Human metabolomics studies reported that reduced SCFAs levels in patients with IBD are associated with a loss of bacterial species that produce SCFAs (33). Treatment with one of the SCFAs producers, *Bacteroides vulgatus* has been shown to increase the abundance of gut microbiota synthesizing SCFAs and relieve DSS-induced mouse colitis (34).

SCFAs can reshape the gut ecology, induce immune modulation, and mediate inflammatory signaling cascade during gut inflammation (35). SCFAs can cross the intestinal epithelial mucosa and be absorbed and utilized by the Smct protein-mediated active transport (36). Moreover, SCFAs also exert anti-inflammatory effects by inhibiting the release of pro-inflammatory cytokines from macrophages and neutrophils (37). In this study, the level of SCFAs in *A. shahii*-treated mouse feces was found to be significantly elevated, primarily acetic and propanoic acids, and the expression of sodium-coupled monocarboxylate transporter gene *Smct* was also increased, likely due to the improvement of SCFAs' utilization. Further, we found a decreased level of IL1β, TNFα, and IL6 in the colon, which may be caused by elevated SCFAs levels.

Bacterial metabolic products that can disrupt or stabilize intestinal epithelial tight junctions and barrier function affect the development of colon inflammation in mice (38). Cecal metabolite profiling and pathway enrichment analysis in this study identified histidine metabolism and pyruvate metabolism pathways were most differentially regulated in colitis mice in comparison with normal mice and *A. shahii* As360-treated mice. Histidine was found to be much higher in the cecal metabolites of colitis mice compared to normal mice. Histamine as a byproduct of histidine metabolism has been reported to aggravate colitis severity in mice (39). However, histamine was not detected in the cecal metabolites in this study. Further studies are required to determine the relationship between histidine metabolism and colitis. Lactic acid as a byproduct of pyruvate metabolism was increased in colitis mice in this study. Lactic acid has been shown to be elevated in some cases of active IBD (40). Furthermore, lactic acid is the end product of anaerobic glycolysis, often associated with inflammation (41). Treatment with *A. shahii* As360 reduced lactic acid in the cecal content of the colitis mice and may prevent inflammation-associated glycolytic shift. However, further studies are needed to

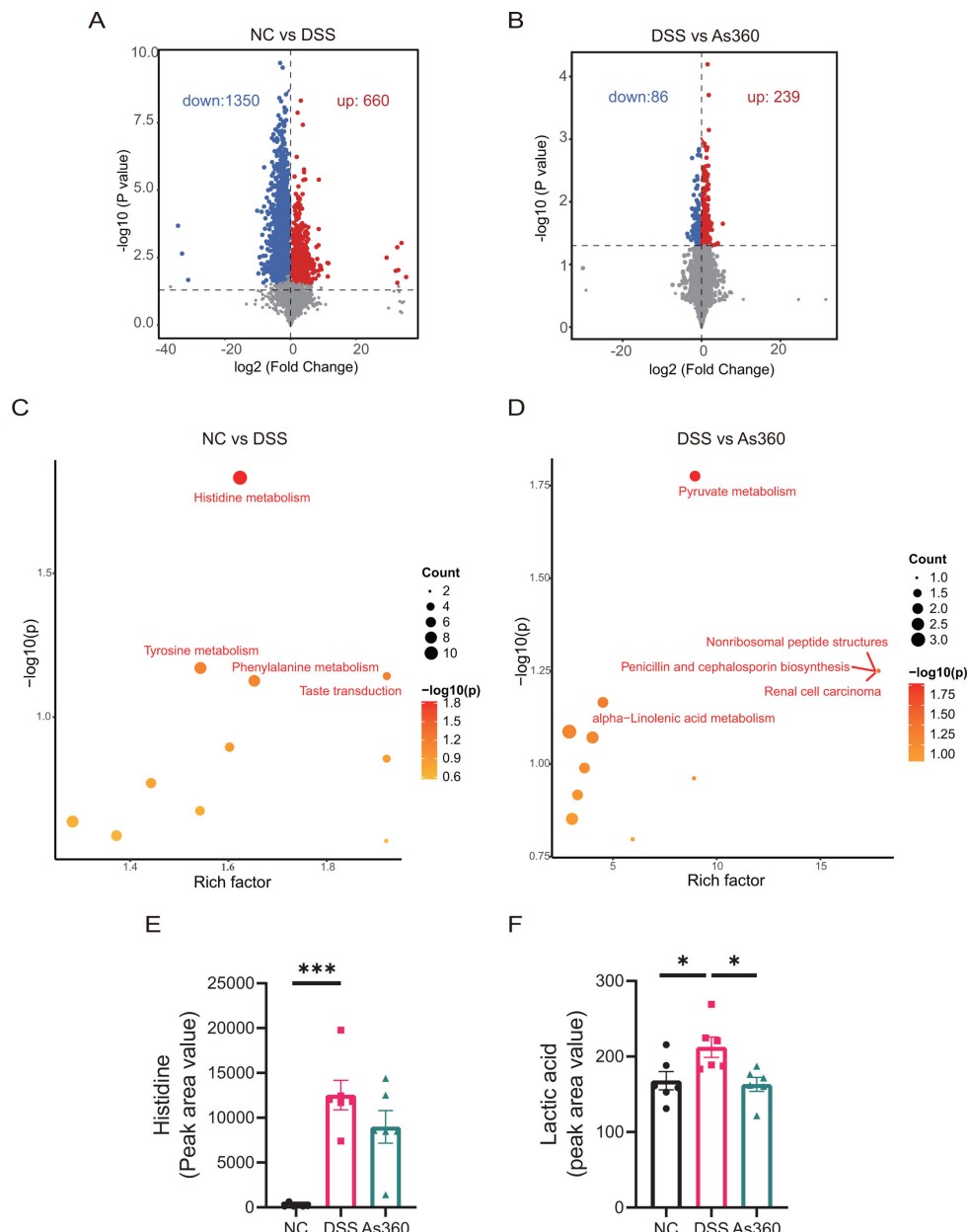

**FIG 7** Analysis of metabolic pathways in mice gut microbiota. (A and B) Volcanic plot of differential metabolites in the cecal contents; the red dots represent an upregulation of the relative abundance of metabolites, the blue ones represent downregulation, and the gray dots represent no significance. The size of dots represents the variable importance plot value (VIP). (C and D) Bubble plot of the differential metabolite KEGG enrichment factor. The scatterplots of enriched KEGG pathways based on significantly altered metabolites. (E and F) The production of histidine and lactic acid. Statistical comparison was performed by a one-way ANOVA followed by Dunnett's multiple comparisons test. Values represent mean ± standard error; number of mice per group, $n = 6$, $*P < 0.05$ and $***P < 0.001$.

elucidate the protective mechanisms of *A. shahii*-mediated metabolites against intestinal barrier damage.

Taken together, the findings from this study suggest that oral administration of the intestinal core microbe *A. shahii* alleviated symptoms of colitis, modulated the release of cellular inflammatory factors, protected the intestinal epithelial barrier, restored gut microbiota disorders, and regulated metabolic levels (Fig. 8). Notably, this study showed that *A. shahii* treatment altered immune responses and improved barrier protection,

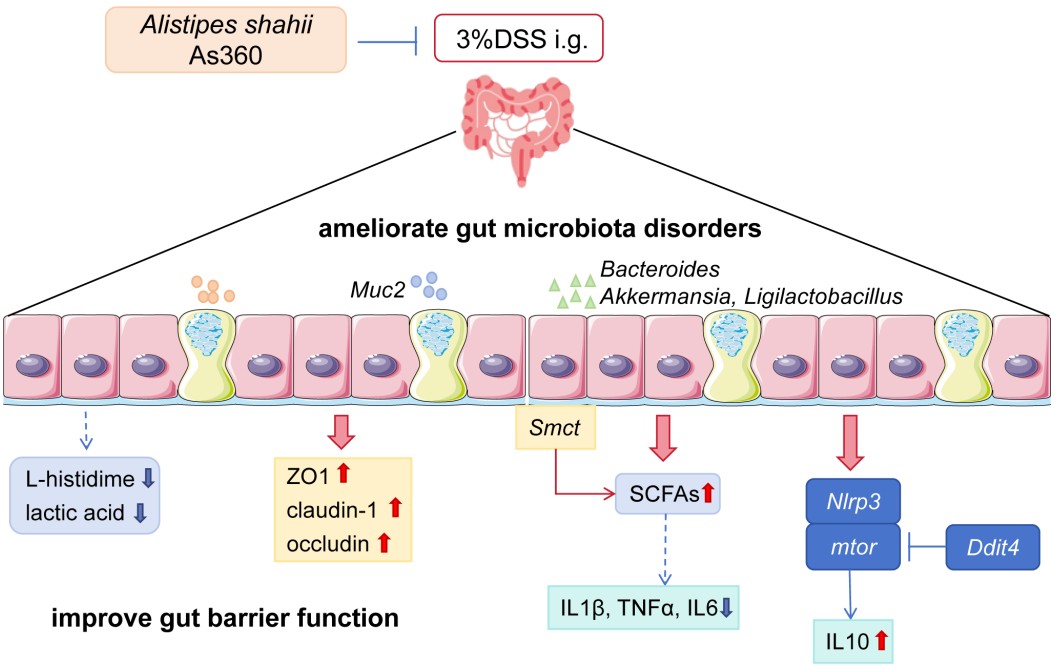

**FIG 8** Schematic representation of anti-inflammatory effect of *A. shahii* As360 on DSS-induced colitis mice. The schematic summarizes the findings of the effect of *A. shahii* As360 on DSS-induced colitis mice using the column epithelium layer to illustrate the changes involved.

but the specific mechanisms involved need to be further studied. Another limitation is that the exact mechanism by which the metabolites regulate intestinal barrier function remains unclear. Nevertheless, our study revealed the beneficial effects of *A. shahii* on colitis mice and our findings provide the basis for future studies and perspectives for better strategies to prevent IBD.

## MATERIALS AND METHODS

### Isolation of *A. shahii* strains from healthy individuals

Fecal samples from healthy volunteers were collected and initially frozen at −80°C. The samples were thawed slowly on ice and diluted in PBS in a gradient ranging from $10^{-1}$ to $10^{-4}$, and 100 µL dilutions were evenly spread on brain heart infusion (BHI; OXOID, USA) agar supplemented with 5% (vol/vol) defibrinated sheep blood (BHI-5% sheep blood). After anaerobic incubation at 37°C for 4 days, gray to opaque circular colonies were selected and further subcultured on BHI-5% sheep blood plate. Purified colonies were maintained in the BHI broth with 20% glycerol and stored at −80°C. For further culturing, isolates were revived from −80°C and cultured in BHI-5% sheep blood plate at 37°C for 48 h in an anaerobic chamber.

The total genomic DNA was extracted using Wizard Genomic DNA Purification kit (Promega, USA) according to the manufacturer's instructions and the 16S rRNA gene amplification product was obtained by using the universal primer 27F/1492R (42), which was sent to RuiBotech Co. for sequencing based on the Illumina platform, and the resulting sequence was uploaded to the NCBI database (GenBank accession numbers: OQ683800, OQ683802, OQ683803, and OQ683805). The nucleotide sequences were compared with the available sequence of *Alistipes* species in the NCBI database using BLAST algorithm for identification.

## Resistance to antibiotics

The antibiotic sensitivity of *A. shahii* isolates was assessed by E-test strips method to determine the minimum inhibitory concentration of various antimicrobial agents, according to the Performance Standards for Antimicrobial Susceptibility Testing made by the Clinical and Laboratory Standards Institute M100 (43). *Bacteroides fragilis* ATCC 25285 was served as quality control. The antibiotics tested were amoxicillin/clavulanate, clindamycin, imipenem, meropenem, metronidazole, penicillin, ampicillin, ceftriaxone, chloramphenicol, and moxifloxacin (Liofilchem, TE, Italy).

## Animal experiments and experimental design

C57BL/6 mice (female, 6 weeks) were obtained from Beijing Charles River Laboratory Animal Technology and the animal experimental protocol was approved by the Welfare & Ethical Inspection in Animal Experimentation Committee at the Chinese CDC (Appl. no.: 2022-021). All mice were housed in a specific pathogen-free environment at 25°C with a 12 h light/dark cycle, *ad libitum* drinking and feeding. After 3 days of climatization, twenty-four mice were randomly divided into the three groups: normal control group, DSS (36–50 kDa, MP Biomedicals, Canada)-induced colitis group, and *A. shahii* As360-treated colitis group. Mice were orally administered 200 µL As360 with $2 \times 10^9$ CFU (As360 group) and equal volume of PBS (DSS group) daily for 14 days. The same gavage regimen was performed with 3% (wt/vol) DSS given in drinking water for the DSS and As360 groups for 0–6 days. The bacterial suspension of *A. shahii* As360 was transported anaerobically using AnaeroGen (Oxoid Ltd.) in anaerobic jars. The body weights of the mice were recorded daily, and fecal properties and occult blood status were added in the last 7 days. The daily DAI, consisted of scoring for weight loss, rectal bleeding, and stool consistency, was scored according to the previous research (44). The feces were collected on day 4 and all mice were euthanized for sample collection in further analysis 2 days after the administration of DSS stopped (45). The length of the colonic tissues was measured from cecum to rectum and photographed. The contents of the cecum were collected, snap-frozen in liquid nitrogen and stored at −80°C.

## Histopathology and immunohistochemistry analysis

The distal colon samples were fixed in 4% paraformaldehyde (Biosharp, Beijing, China), paraffin-embedded, and then stained with hematoxylin and eosin (H&E) by Wuhan Servicebio Technology Co., Ltd. (Wuhan, China). Sections of the same location per mouse were selected to evaluate colon impairment. Histopathological scores were examined in a blinded fashion using previously published criteria by Wang at el. (46).

Immunohistochemical assays were performed based on the above tissue sections. The primary antibodies incubated were claudin-1 and ZO1, respectively, while the secondary antibodies were horseradish peroxidase-labeled for the corresponding species. Aipathwell digital pathology image analysis software was used to automatically analyze protein positivity, described by the positive area (brownish yellow) ratio (47).

## Quantitative real-time polymerase chain reaction

To extract total RNA in colon tissue, the middle portion of colons soaked in RNAlater (Invitrogen, USA) was homogenized with TRIzol reagent (Ambion, Carlsbad, CA, USA) as described previously (48). Then, the RNA was reverse-transcribed to cDNA using the oligo (dT) primers in PrimeScipt RT reagent Kit (TaKaRa, Kusatsu, Japan). Gene amplification was carried out using TB Green Supermix (TaKaRa). Relative expression was calculated based on the 2-ΔΔCT method, and *Gapdh* was used as the internal control. The primer sequences are listed in Table 1.

**TABLE 1** Primer sequences for quantitative real-time polymerase chain reaction

| Gene | Forward (5′–3′) | Reverse (3′–5′) |
|---|---|---|
| *Gapdh* | GCAAGTTCAACGGCACAG | CGCCAGTAGACTCCACGAC |
| *Ddit4* (24) | CTCTGGGATCGTTTCTCGTC | GACACCCCATCCAGGTATGA |
| *mtor* (24) | AAGGCCTGATGGGATTTGGG | GGGGCAGCAGGTTAAGGATT |
| *Nlrp3* | CCGCGTGTTGTCAGGATCTC | AAGGGCATTGCTTCGTAGATAGA |
| *Smct* (49) | ATGCATTCGTCTCTGTGGCA | ATGCATTCGTCTCTGTGGCA |
| *Zo1* | CCTAAGACCTGTAACCATCT | CTGATAGATATCTGGCTCCT |
| Claudin-1 | GCTGGGTTTCATCCTGGCTTCT | CCTGAGCGGTCACGATGTTGTC |
| Occludin | GCCCAGGCTTCTGGATCTATGT | GGGGATCAACCACACAGTAGTGA |
| *Muc2* | GCTGACGAGTGGTTGGTGAATG | GATGAGGTGGCAGACAGGAGAC |

## Serum and colonic inflammatory factor assays

The serum was collected by centrifuging mice blood at 3,000 × *g* for 15 min at 4℃. The proximal colonic tissues were homogenized in buffer containing protease inhibitors (LANBLEAD, China), and then thoroughly ultrasonic ground on ice for 10 s. After centrifugation at 8,000 × *g* for 20 min at 4℃, cell-free supernatant was collected for detection. The concentrations of TNFα, IL10, IL1β, and IL6 in mice colon were measured by enzyme-linked immunosorbent assay (ELISA) kits (R&D Systems, Minneapolis, MN, USA) according to the manufacturer's instructions.

## Inflammatory cytokines in LPS-induced RAW264.7 cells

Assessing the anti-inflammatory effects of *A. shahii* As360 was carried out as described by a previous study with slight modifications (50). Murine macrophage cell line RAW264.7 cells were seeded into 24-well tissue culture plates at a density of $1 \times 10^6$ cells/well in 1 mL Roswell Park Memorial Institute (RPMI) 1640 medium (CORNING, Beijing, China) containing bacterial suspension in the ratio of 1:20 multiplicity of infection, with LPS (1 µg/mL) stimulating simultaneously. Saline with or without LPS-supplemented RPMI 1640 was used as a positive or negative control, respectively. After incubation, culture medium was centrifuged and collected at 2,000 × *g* for 20 min. Cell free supernatants were collected for measurements of four inflammatory cytokines using an ELISA kit (R&D Systems, MN, USA) .

## Microbiome profiling of fecal samples

Fecal bacterial genomic DNA was extracted and the V3–V4 high variant region of 16S rDNA was amplified using 338F/806R primers (34). Sequencing was conducted on an Illumina Novaseq 6000 platform (Biomarker Technologies Co., Ltd., Beijing, China). Low-quality sequencing reads were filtered by Trimmomatic v0.33 (51) and the primer sequences were identified and removed by cutadapt 1.9.1. Amplicon data were denoised using Divisive Amplicon Denoising Algorithm 2 (52) in QIIME2 v2020.6 (53), which was also used to calculate all indices of our samples. USEARCH v10 (54) was used to splice the double-ended reads and remove the chimeras (UCHIME [55], version 8.1), resulting in high-quality sequences, species annotation, and abundance analysis. Alpha diversity was elucidated by Chao1 and ACE. Based on the unweighted UniFrac distance, beta diversity was calculated and described by PCoA. Similarity analysis was performed to test the differences between sample points of different groups. The prediction of bacteria differing between groups was performed based on LefSe and the threshold was LDA > 3.5.

## Fecal SCFAs targeted metabolomics assays

The SCFA standards were accurately weighed to prepare the mixed standard linear masterbatch, which was diluted by methanol to obtain a series of SCFA calibrators.

Certain concentrations of isotope standards were mixed as internal standards. The samples (100 µL) were homogenized with 500 µL of 80% methanol and centrifuged to remove the protein. The supernatant was added to derivatization reagent (100 µL) and derivatized at 40°C for 40 min. Then supernatant (100 µL) was homogenized with 1 µL mixed internal standard solution to be injected into the liquid chromatography coupled to tandem mass spectrometry (LC-MS/MS) system for analysis.

An ultra-high performance liquid chromatography (UHPLC)-MS/MS system (Vanquish Flex UHPLC-TSQ Altis, Thermo Scientific, Germany) was used to quantitate SCFAs in Novogene Co., Ltd. (Beijing, China). Separation was performed on a Waters ACQUITY UPLC BEH C18 column (2.1 × 100 mm$^2$, 1.7 µm) which was maintained at 40°C. The mobile phase, consisting of 10 mmol/L ammonium acetate in water (solvent A) and acetonitrile:isopropanol (1:1) (solvent B), was delivered at a flow rate of 0.30 mL/min. The solvent gradient was set as follows: initial 25% B, 2.5 min; 25–30% B, 3 min; 30–35% B, 3.5 min; 35–38% B, 4 min; 38–40% B, 4.5 min; 40–45% B, 5 min; 45–50% B, 5.5 min; 50–55% B, 6.5 min; 55–58% B, 7 min; 58–70% B, 7.5 min; 70–100% B, 7.8 min; 100–25% B, 10.1 min; and 25% B, 12 min. The mass spectrometer was operated in negative multiple reaction mode. Parameters were as follows: ion spray voltage (−4,500 V), sheath gas (35 psi), ion source temp (550°C), auxiliary gas (50 psi), and collision gas (55 psi).

## Quantification of SCFAs in culture supernatant

The SCFAs in the fermentation supernatant samples of the strains were extracted with methyl tert-butyl ether and analyzed using a SHIMADZU GC2030-QP2020 NX gas chromatography-mass spectrometer with Agilent HP-FFAP capillary tubes (56). The concentrations of acetic, propionic, isobutyric, butyric, isovaleric, valeric and hexanoic acids were determined.

## Untargeted metabolomic profiling of cecal contents samples

The LC/MS system for metabolomics analysis is composed of Waters Acquity I-Class PLUS ultra-high-performance liquid tandem Waters Xevo G2-XS QT of high resolution mass spectrometer. The column used was purchased from Waters Acquity UPLC HSS T3 column (1.8 µm 2.1*100 mm$^2$). For positive/negative ion mode, mobile phase A was 0.1% formic acid aqueous solution; mobile phase B was 0.1% formic acid acetonitrile; and injection volume was 1 µL. Based on the acquisition software (MassLynx V4.2, Waters) in MSe mode for dual-channel data acquisition, the original data were processed by Progenesis QI software for peak extraction. Peak pairing and other data processing operations were based on Progenesis QI software online METLIN database and Biomark's self-built library for identification, while theoretical fragment identification and quality deviation are within 100 ppm.

The classification and pathway information were searched in HMDB (57) and LIPID MAPS (58) database, and the difference multitimes were calculated and compared according to the group information. The variable importance plot (VIP) value of the model was calculated using multiple cross-validation, and the differential multiple $P$ value of the OPLS-DA model was combined with the VIP value to screen for differential metabolites. The screening standard was FC > 2 or < 0.5, $P$ value < 0.05 and VIP > 1, which was calculated to enrich the significant differential metabolites by KEGG pathway using hypergeometric distribution test (59).

## Statistical analysis

Statistical analysis was performed using GraphPad Prism v9.0.0. The experimental data were compared using one-way ANOVA with Dunnett's multiple comparison test. Spearman correlation analysis was conducted to determine the repeatability of samples and quality control samples in the group. Pairwise comparisons of each compound in untargeted metabolomic profiling were performed using Welch's test. The R package,

ropls, was used to conduct OPLS-DA modeling, and 200 replacement tests were done to verify the reliability of the model. $P < 0.05$ denotes statistically significant.

## ACKNOWLEDGMENTS

This work was supported by the Discovery and function of unknown microorganisms (No. 2018RU010).

## AUTHOR AFFILIATIONS

[1]School of Public Health, Nanjing Medical University, Nanjing, China

[2]National Key Laboratory of Intelligent Tracking and Forecasting for Infectious Diseases, National Institute for Communicable Disease Control and Prevention, Chinese Center for Disease Control and Prevention, Beijing, China

[3]Department of Epidemiology and Statistics, School of Public Health, Hebei Medical University, Shijiazhuang, Hebei, China

[4]Hebei Key Laboratory of Environment and Human Health, Shijiazhuang, China

[5]School of Biotechnology and Biomolecular Sciences, University of New South Wales, Sydney, New South Wales, Australia

[6]Research Units of Discovery of Unknown Bacteria and Function, Chinese Academy of Medical Sciences, Beijing, China

[7]Hebei Key Laboratory of Intractable Pathogens, Shijiazhuang Center for Disease Control and Prevention, Shijiazhuang, China

## AUTHOR ORCIDs

Xiaoying Lin http://orcid.org/0000-0003-0063-070X
Ruiting Lan http://orcid.org/0000-0001-9834-5258
Liyun Liu http://orcid.org/0000-0003-2257-0277

## AUTHOR CONTRIBUTIONS

Xiaoying Lin, Data curation, Formal analysis, Methodology, Writing – original draft, Writing – review and editing | Mingchao Xu, Methodology, Writing – review and editing | Ruiting Lan, Writing – review and editing | Dalong Hu, Writing – review and editing | Suping Zhang, Methodology | Shuwei Zhang, Methodology | Yao Lu, Methodology | Hui Sun, Methodology | Jing Yang, Investigation, Project administration | Liyun Liu, Funding acquisition, Project administration, Writing – review and editing | Jianguo Xu, Funding acquisition, Project administration, Resources

## DATA AVAILABILITY

The 16S rRNA amplicon sequencing data used in this study are publicly available in the NCBI Sequence Read Archive (SRA) database (accession number PRJNA1119725). *Alistipes shahii* 16S rRNA sequences were uploaded to the NCBI database (GenBank accession numbers: OQ683800, OQ683802, OQ683803, and OQ683805).

## ADDITIONAL FILES

The following material is available online.

### Supplemental Material

**Figure S1 (mSystems01607-24-s0001.eps).** Weight change for 7 days in the pre-gavage stage.
**Figure S2 (mSystems01607-24-s0002.eps).** Rarefaction curves.
**Supplemental material (mSystems01607-24-s0003.docx).** Legends for Figures S1 and S2; Tables S1 and S2.

## Open Peer Review

**PEER REVIEW HISTORY (review-history.pdf).** An accounting of the reviewer comments and feedback.

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
