## [Reviewer comments · mSystems]

Gut commensal *Alistipes shahii* improves experimental colitis in mice with reduced intestinal epithelial damage and cytokine secretion

Xiaoying Lin, Mingchao Xu, Ruiting Lan, Dalong Hu, Suping Zhang, Shuwei Zhang, Yao Lu, Hui Sun, Jing Yang, Jianguo Xu, and Liyun Liu

Corresponding Author(s): Liyun Liu, National Institute for Communicable Disease Control and Prevention

Review Timeline:

Submission Date:

December 2, 2024

Accepted:

January 6, 2025

Editor: Hiutung Chu

Reviewer(s): Disclosure of reviewer identity is with reference to reviewer comments included in decision letter(s). The following individuals involved in review of your submission have agreed to reveal their identity: Shuai Hao (Reviewer #1); Ruifu Yang (Reviewer #4)

Transaction Report:

DOI: <https://doi.org/10.1128/msystems.01607-24>

Re: mSystems01607-24 (**Gut commensal *Alistipes shahii* improves experimental colitis in mice with reduced intestinal epithelial damage and cytokine secretion**)

Dear Dr. Liyun Liu:

Your manuscript has been accepted, and I am forwarding it to the ASM production staff for publication. Your paper will first be checked to make sure all elements meet the technical requirements. ASM staff will contact you if anything needs to be revised before copyediting and production can begin. Otherwise, you will be notified when your proofs are ready to be viewed.

Sincerely,
Hiutung Chu

Editor
mSystems

Reviewer #1 (Comments for the Author):

The author has revised the article in accordance with the requirements, Thank you.

Reviewer #4 (Comments for the Author):

The MS has been adequately revised according to the reviewers' comments. I have no further questions about this MS.